# Rapid Changes to Endomembrane System of Infected Root Nodule Cells to Adapt to Unusual Lifestyle

**DOI:** 10.3390/ijms24054647

**Published:** 2023-02-28

**Authors:** Elena E. Fedorova

**Affiliations:** Timiryazev Institute of Plant Physiology, Russian Academy of Science, 127276 Moscow, Russia; elenafedorova06@mail.ru

**Keywords:** symbiosis, root nodule, cell membranes, membrane transporters, intracellular bacteria accommodation

## Abstract

Symbiosis between leguminous plants and soil bacteria rhizobia is a refined type of plant–microbial interaction that has a great importance to the global balance of nitrogen. The reduction of atmospheric nitrogen takes place in infected cells of a root nodule that serves as a temporary shelter for thousands of living bacteria, which, per se, is an unusual state of a eukaryotic cell. One of the most striking features of an infected cell is the drastic changes in the endomembrane system that occur after the entrance of bacteria to the host cell symplast. Mechanisms for maintaining intracellular bacterial colony represent an important part of symbiosis that have still not been sufficiently clarified. This review focuses on the changes that occur in an endomembrane system of infected cells and on the putative mechanisms of infected cell adaptation to its unusual lifestyle.

## 1. Introduction

Symbiosis between leguminous plants and soil bacteria rhizobia is initiated by a signal exchange between the host plant and bacteria. In response to flavonoids secreted by legume roots, rhizobia synthesize lipochito-oligosaccharids, which are nodulation factors (Nod factors) that initiate the expression of genes of the so-called Nod factor signaling pathway, thereby inducing nodule organogenesis. The Nod factor signaling pathway has been extensively studied and reviewed, and detailed information concerning this early stage of symbiosis can be found in several excellent reviews [1,2,3,4].

However, the functional role of the root nodule is associated with later stages of symbiosis when the colony of bacteria accommodated in a living plant cell become capable of reducing atmospheric nitrogen. The aim of this paper was to review the data concerning the changes in a host cell endomembrane system that occur during the short but quite important co-existence of symbiotic partners and to specify possible directions for new research.

## 2. Root Nodules

Nitrogen-fixing root nodules are transitory organs formed on plant roots upon inoculation by symbiotic microorganisms. Legume root nodule development starts with the initiation of a new meristem from the dedifferentiated root cortical cell followed by root nodule organogenesis [5,6,7,8]. The spatial pattern of meristem initiation determines the anatomical pattern and the growth of root nodules [5]. In a nodule with a meristem situated at the nodule apex, newly produced postmeristematic cells are shifted downward with a basipetal gradient of cell differentiation. Such a nodule develops as an elongated cylindrical structure and is termed the indeterminate type of growth nodule, examples of which are nodules of *Medicago truncatula* and *Pisum sativum*. The activity of the apical meristem in these nodules persists for 5–6 weeks [1,5,9,10,11]. Nodules with apical and lateral meristematic loci develop in globoid form with a centripetal developmental gradient, examples of which are nodules of *Phaseolus vulgaris*, *Lotus japonicum,* and *Glycine max*. The oldest cells are situated in the center of the nodule and are covered by layers of younger cells [6,12]. This type of nodule is termed the determinate type of growth nodule because its growth mostly depends on cell enlargement [6]. The lupinoid nodules of *Lupinus albus* and *Lupinus luteus* are globoid in form but contain lobes with an indeterminate type of growth as well as the infected cells, which are able to divide [13]. Several excellent reviews dedicated to nodule organogenesis have addressed the details [5,6,14,15,16].

### 2.1. From Apoplastic Bacteria to Symbiosomes

At the initial stages of nodule formation, rhizobia enter the intercellular space of nodule primordia that is initiated on the young root. Bacteria are unable to enter the symplast of mature cells, probably due to the developed cell wall and a high turgor pressure; however, infected threads carrying rhizobia enter the symplast of young, postmeristematic cells [5,10]. Inside the host cell, the microsymbionts are situated in specific asymmetric protrusions of the plasma membrane. In legume root nodules, such protrusions are tubular structures called infection threads, the extensions of infection threads with reduced cell walls are termed infection droplets, and released bacteria surrounded by a host cell-derived membrane are called symbiosomes (Figure 1) [17,18,19]. Root nodule development is accompanied by a massive transcriptional reprogramming that causes changes in the anatomy, physiology, transcriptome, and metabolome of host cells [4,19,20]. Infected cells maintain thousands of living rhizobia for a prolonged period of up to 6–7 weeks; hence, they are permissive for intracellular accommodation of bacteria and must be considered as special biological units—symbiotic cells. These cells are protected from the default answers for infection such as programmed cell death (PCD) by a set of genes that causes the suppression of innate immunity responses [20,21,22,23,24,25]. However, this suppression is not universal, and some antimicrobial peptides (NCRs) that are synthesized in legumes from the inverted repeat-lacking clade (IRLC) including *M. truncatula* cause the terminal differentiation of intracellular bacteria or the termination of incompatible symbiosis [26,27]. The specific environment that is maintained in infected cells helps (or forces) intracellular rhizobia at a certain stage of development to start the reduction of nitrogen from the air via the bacterial enzyme nitrogenase. Mature symbiotic cells are partly hypoxic, which allows the functional activity of oxygen-sensitive nitrogenase [28].

The rhizobia that have entered the symplast of the host cell are kept separate from the host cytoplasm by the membrane, the source of which initially is the plasma membrane of the host cell [17,18]. The symbiosome has some structural analogy with pathogenic vacuoles that house microbes in mammals [29]. The bacterial pathogens *Salmonella*, *Mycobacteria*, *Legionella*, *Chlamydia*, and *Brucella* temporarily reside in membrane compartments—bacteria-containing vacuoles. The modification of endocytic, exocytic, and/or ER-to-Golgi vesicle trafficking of invaded cells helps to maintain the bacterial population [29,30]. The pathogenic vacuoles mostly are destined to fuse with lysosomes of the host cell with a consequent lytic clearance of bacteria. In infected cells of root nodules, however, most symbiosomes persist as individual units for 3–4 weeks and do not fuse with the host cell vacuoles [31,32]. Some putative mechanisms that inhibit fusion with the host vacuole in infected cells have been described for *M. truncatula* root nodules [32,33]; these include a gradual change in the identity of the symbiosome membrane as well as changes in the functionality of the vacuole of the infected cell. At the same time, lysis of bacteria and termination of symbiosis is induced in many cases that include incompatible interactions and environmental stresses [27,34,35]. The comprehensive model of the processes that prevent lytic clearance of bacteria has not yet been developed.

### 2.2. Symbiosome Identity and Symbiosome Membrane

All membrane-bound organelles of eukaryotic cells have their own identity that defines the compartment-depending membrane features [36]. The identity of cell organelles is determined by specific membrane-bound proteins that are mostly involved in the process of membrane fusion. These are the regulatory GTPases of the Rab family and the Rab-interacting integral proteins of the soluble NSF attachment protein receptor (SNARE) families [36,37]. During the fusion of vesicles with the membrane, matching types of SNAREs form a highly stable protein association called the SNARE complex [37,38]. Studies in plants have shown that most SNAREs are associated with specific intracellular compartments [39,40]. Some animal intracellular pathogens utilize the strategy of host cell mimicry; for example, *Salmonella*-containing vacuoles acquire Rab proteins and hence accept an identity similar to early/late endosomes [28,29].

Symbiosomes in root nodules of *M. truncatula* acquire a specific set of membrane identity proteins, which is similar to other cell organelles. This event determines membrane traffic as well as the dynamic changes in symbiosome membrane features (Figure 1). Rhizobia cells that have entered the host cell have a fragment of the plasma membrane as an integral part of the symbiosome membrane, and symbiosomes of *M. truncatula* and soybean contain plasma-membrane Syntaxin 134 and vesicular v-SNARE VAMP72, the markers of plasma-membrane-targeted exocytosis [31,32]. However, the “classical” markers of endocytosis (small GTPase Rab5 and trans-Golgi network identity marker SYP4) do not show an association with freshly released rhizobia, therefore the “endocytotic” entrance of rhizobia into the host cell has some pronounced deviations [31].

The molecular marker of endocytosis—small GTPase Rab7—is temporarily present on the membrane of endosomes and on tonoplast; this marker appears on the symbiosome membrane at later stages of development [12,31,32]. Mature symbiosomes also recruit tonoplast-resident vacuolar SNAREs from the so-called vacuolar SNAREpin complex [31,32,41]. Evidently, mature symbiosomes have a mosaic identity and combine different markers of the plasma membrane and the endosome/vacuole on their membranes (Figure 1). It can be assumed that anterograde, retrograde, and endocytic trafficking pathways may be operational in transport toward the symbiotic membrane. The acceptance of host cell molecular markers may facilitate retargeting of membrane transporters (for example, aquaporins) toward the membrane, thereby ensuring the growth and development of the symbiosome [42,43].

The host cell membrane transporters that have been identified on the symbiosome membrane include the iron transporter [44,45], the iron-activated citrate transporter [46], the molybdenum transporter [47], Zinc-Iron Permease6 [48], the SST1 sulfate transporter [49], copper transporter1 [50], peptide transporter Soybean Yellow Stripe-like 7 [51], and several putative candidates for malate transporters [52]. The recently presented work by Luo et al. [53] described a quantitative proteomics analysis of soybean root nodule symbiosome membranes and provided a framework of putative research in transport toward the symbiosome membrane and the regulatory mechanisms of this transport.

### 2.3. Membrane Sources for Host Cell and Microsymbiont Expansion

During nodule development, the complete maturation of symbiosomes of *M. truncatula* requires 5–7 cell layers, with the most rapid growth in 1–2 cell layers proximal to the zone of nitrogen fixation in interzone 2–3 [11]. The infected cells grow concomitantly with the increasing symbiosome population. As a result, the infected cell’s volume increases several times in comparison with that of a non-infected cell [33]. This speedy expansion brings under consideration the putative membrane resources provided by the host cell for this process as well as the causes that force the host to maintain such explosive growth. The most obvious membrane resource is an exocytotic pathway with post-Golgi vesicles [12,17,31,54] and endoplasmic reticulum (ER), which has long been considered to be one of the sources of the membrane for symbiosomes [17]. The ER is always abundant in young infected cells, and the contacts of ER vesicles with symbiosome membranes have been displayed by using electron microscopy [12,17,31,54,55].

The rapid growth of infected cells and the propagation of intracellular bacteria raises a question regarding the causes of the synthesis of membranes by the host cell at such an unprecedented scale. The plasma membrane is known to be inelastic and unable to stretch more than 3% [56], hence the growth of new membrane interfaces as the plasma membrane or the membrane enveloping an infection thread, unwalled droplet, or symbiosome depend on the host cell resources. The growth of symbiotic structures in infected cells is mainly isodiametric for unwalled droplets and symbiosomes (apart from a tip growth of infection threads). Hence, the membrane resources have to be targeted to support the expansion indiscriminately of the growth type. The pronounced changes in membrane interphase in an infected cell include ER–plasma membrane remodeling and changes in membrane curvature, similar to a recent report by Rosado and Bayer [57]. At the same time, the idea that the ER is just a source of membrane for expanding symbiosomes is a very simplified view of a very complicated matter. Apart from the available membrane, the ER is a source of a plethora of different molecules. It is difficult to estimate the number of proteins that may be excreted in this manner into the symbiosome space at different time points of this communication.

The inner space of organelles of symbiotic origin (mitochondria and plastids), according to Bellucci et al. [58], can be defined as an “external space” that is similar to the apoplast. It is quite plausible that the symbiosome also can be defined as “the extracellular space”. The recent data reported by Luo et al. [53] provided some support for this speculation. According to the data obtained via label-free quantitative proteomic technology for the soybean symbiosome membrane (SM), peribacteroid space (PBS), and root microsomal fraction (RMF), material exchange and signal communication indicate the likely extracellular nature of the symbiosome [53]. Several coatomer proteins of the COPI complex were detected in the SM proteome; the COPI and COPII complexes are known to be involved in retrograde and anterograde trafficking between the ER and Golgi apparatus [53]. Secretory proteins released into the ER lumen can be retained in this compartment, move along the secretory pathway, be transported to vacuoles in plants (bypassing the Golgi apparatus), or be excreted to the external (extracellular) space [58]. We can guess that in infected cells, the transport via ER (which is yet unexplored), is an important method of communication between symbiotic partners.

The retargeting toward the symbiotic interface of membrane resources that were pre-directed to the plasma membrane, endosomes, and tonoplast favors the propagation of the intracellular microsymbiont colony [12,18,59]. Concerning the change in the destination of these membrane resources, it was proposed that the membrane tension created by the expanding microsymbiont provides the vector for targeted endomembrane traffic toward the new forming membranes in infected cells of *M. truncatula* [59]. Protein trafficking via the endomembrane system is tightly regulated in response to environment stimuli [60,61], but the response of the cells to mechanical stress involves quick indiscriminative retargeting of all membrane resources available in place to prevent a possible rupture of the membrane [62]. Hence, the unprecedented increase in membrane formation in the infected cell may be a repair mechanism induced as a response to the membrane stretching caused by the propagation of the microsymbiont in an infected cell. How specific in this case will protein traffic to interface with symbiotic membranes be? Currently, the mechanisms of a specific sorting of proteins toward the symbiotic interphase (if they exist) have not been elucidated.

### 2.4. Host Cell Architecture Remodeling and Cytoskeleton Rearrangement

The mechanisms that adapt the host cell architecture to accommodate intracellular bacteria are not yet clear; however, the host cell cytoskeleton seems to reorganize after bacteria enter the host cell. Actin microfilament rearrangement, which is linked to the positioning of organelles and influences cell shape, provides a roadway for the transport of membrane vesicles [63]. During symbiosome growth in root nodules of *M. truncatula*, the actin microfilament network plays an indispensable role in membrane traffic and the reformation of cytoplasm architecture for symbiosome accommodation [64,65,66,67].

Actin configuration in infected nodule cells changes markedly during infected cell development. Zhang et al. [67] defined five zones in the infected zone of *M. truncatula* root nodules with specific rearrangements of actin. The cytoskeletal patterns are mediated by diverse actin-binding proteins such as actin depolymerization factors (ADFs) [68], formin [64,65,66], Phospholipase Dβ [69] and the ARP2/3 complex [70], which nucleates new actin filaments and forms branched actin networks. The manipulation of host cell actin via the ARP2/3 actin nucleating complex is also used as common strategy for the establishment of an intracellular lifestyle by enteropathogenic bacteria in animal cells [71]. In *Arabidopsis thaliana*, ARP2/3 is strongly associated with cell membranes of the microsomal fraction from several organelles that include the endoplasmic reticulum (ER), tonoplast, plasma membrane, and the early endosome [72,73].

The microtubule cytoskeleton pattern also undergoes changes during symbiosis development [13] starting from reorganization of microtubules on initial stages of root hair responses to rhizobia [74]. In nodule cells, bacteroid positioning correlates with characteristic microtubule rearrangements [75], wherein the pattern in root nodules was found to be host-plant specific [75]. The nodulation-specific kinesin-like calmodulin-binding protein (nKCBP), which crosslinks microtubules with the actin cytoskeleton, controls central vacuole morphogenesis in symbiotic cells in *M. truncatula* [76].

Apart from non-dividing infected cells of *M. truncatula,* the rearrangement of the cytoskeleton in infected cells of *Lupinus albus* is rather unusual. The infected cells of *L. albus* root nodules are able to divide while already infected. The pattern of the cytoskeleton during infected cell mitosis is comparable to that of the other dividing cells. The clustered symbiosomes move to the cell poles during spindle elongation in a manner similar to other host cell organelles. This implies the existence of functional mechanisms of microtubule and microfilament attachment on symbiosome membranes and the presence of suitable identity markers as well as the physical anchoring mechanism [13].

### 2.5. The Game of Volumes: How to Create the Space for Expanding Microsymbionts Inside of an Infected Cell

Throughout their development, plants balance cell surface area and volume with water and ion transport and turgor. This balance lies at the core of cellular homeostatic networks and is central to a plant cell’s capacity to withstand abiotic as well as biotic stress [77].

During the symbiosis development, the volume and surfaces of the host cell and microsymbiont change quite dramatically: the host cell volume increases 5 times, and the volume of intracellular bacteria increases 10 times [33]. How is the space for the microsymbiont created in the infected cell? When symbiosomes of *M. truncatula* nodules begin to fix nitrogen, a developmental switch occurs and concomitant changes in vacuoles features are induced. In infected cells, the expression of VPS11 and VPS39, which are genes of the HOPS complex that are essential to the specific tethering and fusion of vesicles to vacuoles [78], become suppressed. Vacuoles lose their acidic pH, which coincides with the rapid expansion of symbiosomes in the cell layer proximal to interzone 2/3 and the contraction of vacuoles [33] (Figure 2). The tonoplast aquaporin TIP1g is retargeted to the symbiosome membrane, which provides the microsymbiont with an advantage in water transport from the host cell to the bacteroid. As a result of these changes, the total vacuole volume in mature infected cells remains only 30% of total cell volume, whereas for plant cells it is normally from 70 to 90%. At the same time, the volume of symbiosomes increases to occupy 65% of the cell volume [33].

Taking into account the fact that the vacuole is a vital organelle of the plant cell [79], the suppression of the HOPS complex and the loss in vacuole lytic properties in living cells may be ruinous to the host cell’s wellbeing. This situation may be one of the reasons for the extremely short lifespan of symbiotic cells. On the other hand, the vacuole defunctionalization most likely contributes to the maintenance of symbiosomes as individual units, thereby preventing fusion with the lytic compartment [33,80]. As a result, symbiosomes with a “fake” identity of “late endosomes/young vacuoles” due to the presence of endocytotic identical markers on the membrane are able to survive and fix atmospheric nitrogen for several days (the time to fulfill the task for the infected cell).

However, the mechanisms that regulate vacuole defunctionalization, the putative translational regulation of involved proteins [81], and the formation of an osmotic upshift of the symbiosome to ensure the vector of water transport have not yet been clarified.

## 3. Cost

Methods to estimate the cost of nitrogen fixation have been proposed and are usually based on the amount of energy in terms of carbohydrates per the unit of fixed nitrogen [82]. However, the cost of co-existence of symbionts in terms of host cell fitness and life span has ever been considered. Generally, symbiosis is described as mutually beneficial to both partners [83], but the short life span and the inevitable scenario of infected cell death after 4–5 weeks of existence has been well described and accepted as the norm [84,85]. It is a reasonable assumption that the maintenance of an intracellular bacterial colony may have a detrimental effect on the host cell’s wellbeing.

To identify the putative negative effects caused by an intracellular bacterial colony, the well-known fact of the high sensitivity of root nodule to ionic stresses such as salinity was used. Such a low tolerance suggested some pre-existing irregularities in the uptake or distribution of some ions in the nodule tissue [86,87].

As it turned out, the mature nitrogen-fixing cells during the 2–3 weeks of their existence were rapidly losing potassium ions (K^+^) [86,87]. It is understandable that the pool of K^+^ in infected cells is shared between two partners—the host plant cell and several thousand bacteria that propagate and reside inside host cells. This situation significantly increases the demand for potassium (K^+^). However, the deficiency is also caused by the insufficient of K^+^ inward transport. Analysis of the distributions of ion transporter proteins in infected cells versus non-infected ones showed that in mature infected cells, the K^+^ channel MtAKT1 (Figure 3A) and Na^+^/K^+^ exchanger MtNHX7 (Figure 3B) were mistargeted, rapidly shed from their destination membranes, and expelled to young vacuolar compartments. In uninfected cells located nearby, however, the potassium level and the distribution of transporter proteins on the target membranes did not undergo such drastic changes. The application of salt stress revealed the accumulation of 5–6 times more Na^+^ per infected cell versus a non-infected one. Thus, the presence of a bacterial colony in the cell caused the deterioration in the ion exchange and a failure in the distribution of proteins responsible for ion transport [86,87]. It is plausible that the hypoxic conditions of infected cells that promote the nitrogenase activity as well as the alteration in endomembrane traffic in infected cells may be causal in such a situation. Hence, the high sensitivity toward salt stress seems to be a consequence of the infected cell’s environment; i.e., it is a consequence of the presence on bacteria in cell’s symplast. Therefore, this is a default situation for infected cells.

It is understandable that membrane protein biogenesis and proper targeting can fail at any of the various steps: translation, targeting, insertion, folding, or assembly [88]. Currently, we cannot diagnose all mechanisms for dislocation and retargeting of proteins in infected cells, but we can be sure that this situation is widespread and on a much larger scale than mere changes in the distribution of several ion transporters [12,31,32,33,86,87]. This must be taken into consideration in research aimed at improving the stress tolerance of root nodules. Putative gene modification to obtain the overexpression of selected genes as such may be useless if the proteins are not on their correct destination membranes. The real distribution of proteins (including transporters of necessary elements in infected cells) must be verified.

Currently, membrane protein targeting, insertion into the transmembrane domains, translocation, additional folding steps in the membrane, and assembly with interaction partners have not yet been sufficiently studied in an infected cell environment.

Taking into account the data described above, the aim to improve the health of infected cells may represent one of the possible ways to extend the period of active nitrogen fixation and reduce the cost of symbiosis for the host plant.

## 4. Termination

The lifespan of symbiosomes is terminated in the so-called senescent zone [64,84,85]. The typical cytological events during an infected cell’s senescence/symbiosis termination have been well studied and described [84,85]. Symbiosomes in the zone of senescence fuse together, which is followed by the formation of lytic vacuole-like structures [89] as well as the expression of lytical enzymes [90,91] and induction of reactive oxygen species (ROS) [35,92]. Recently, with the aim of clarifying the mechanisms of nodule senescence, Sauviac et al. [93] performed a dual plant–bacteria RNA sequencing approach on *M. truncatula–Sinorhizobium meliloti* nodules that created a comprehensive resource of hundreds of host plant and bacterial genes that are differentially regulated during this stage of symbiosis. Some changes in gene expression in the zone of natural nodule senescence may also be a consequence of intracellular bacterial colony presence in the host cell. It can be assumed that the conditions in an infected cell become suboptimal during its lifespan. Currently, the reasons why an infected cell dies after a maximum of 5 weeks of existence while a genetically identical uninfected cell can live for years have not been explained.

## 5. Conclusions

The changes in endomembranes of infected cells occur at each stage of symbiosis development, beginning with the architecture of the root hair after the contact with rhizobia up to the formation of lytic compartments in the zone of symbiosis termination.

The entrance of microsymbionts triggers the formation of specific asymmetric protrusions of the plasma membrane. The growth of symbiotic structures in infected cells is mainly isodiametric for unwalled droplets and symbiosomes (apart from a tip growth of infection threads).

The special ecological niche for living nitrogen-fixing bacteria is based on the specific membrane interface. Due to the presence of endocytotic identical markers on the membrane, symbiosomes are able to survive and fix atmospheric nitrogen for several days (the time to fulfill the task for the infected cell).

The spectrum of proteins that appear on this and other membranes of an infected cell may not be identical to the spectrum and state of the homologous membrane in uninfected cells from nodules or other tissues. The functionality of proteins and their residence time on membranes in infected cells may have specific differences that distinguish them from proteins in uninfected cells. In this regard, the study of endoplasmic reticulum proteins and their transport into symbiosomes is quite important and timely.

The maintenance of the bacteria colony in the host cell has its price that is expressed in the shortening of the life span, which may be an integral result of the infected cell’s conditions.

The features of an infected cell’s endomembrane system must be taken into consideration in works aimed to improve certain aspects of the symbiotic relationship via creation of genetically modified mutants with an altered level of expression of certain genes.

During the last 20 years, the study of symbiotic relations in plants has made a great progress. We can point to the tremendous advances in research on signaling between partners in the early stages of symbiosis and the excellent transcriptomics research that has made it possible to understand changes in gene expression during symbiosis development. However, a great number of the processes that relate to the regulation of membrane transport, protein targeting, changes in the cytoskeleton, and the regulation of osmotic processes in infected cells are still waiting to be explored.

## Figures and Tables

**Figure 1 ijms-24-04647-f001:**
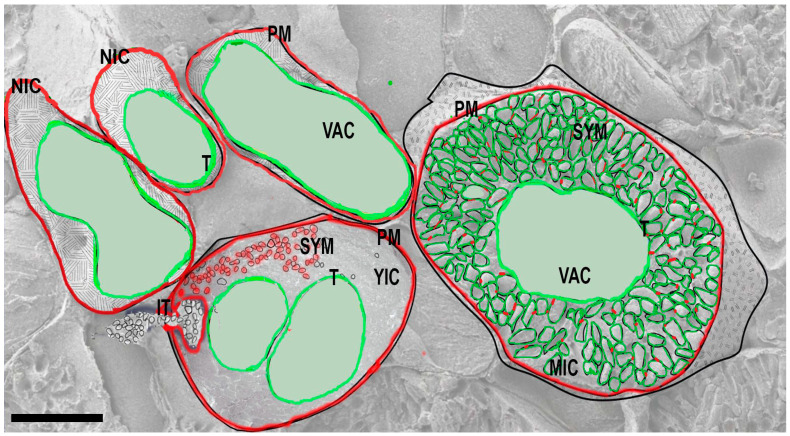
Thechange in the symbiosome membrane’s identity during the development and maturation of infected cells. The membrane of the symbiosome in a young infected cell (YIC) contains a fragment of the plasma membrane (PM, marked in red) as an integral part and contains markers of plasma-membrane-targeted exocytosis. In a mature infected cell (MIC), symbiosomes accept tonoplast-resident vacuolar SNAREs and markers of endosomes (marked in green) but also retain some plasma-membrane markers (red dots). Abbreviations: NIC—non-infected cell, YIC—young infected cell, MIC—mature infected cell, VAC—vacuole, T—tonoplast, PM—plasma membrane, IT—infection thread, SYM—symbiosomes. Bar: 12.5 μm.

**Figure 2 ijms-24-04647-f002:**
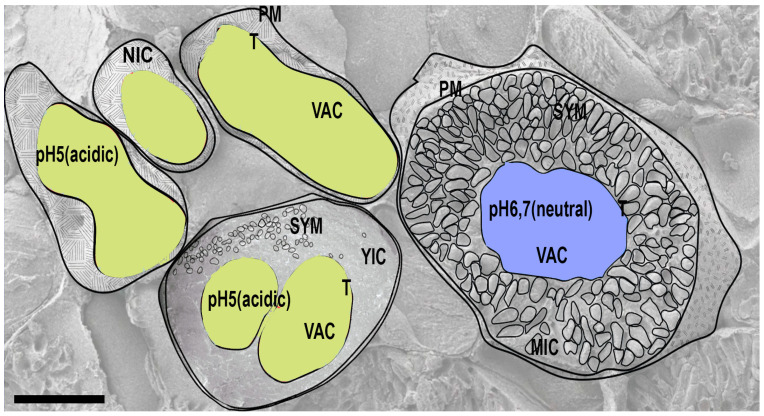
Dynamic changes invacuole pH in non-infected (Non-inf), young infected (YIC), and mature infected (MIC) cells. The vacuoles of mature infected cells lose their acidic pH (marked in yellow) and acquire a neutral pH (marked in blue). Abbreviations: NIC—non-infected cell, YIC—young infected cell, MIC—mature infected cell, VAC—vacuole, T—tonoplast, PM—plasma membrane, SYM—symbiosomes. Bar: 12.5 μm.

**Figure 3 ijms-24-04647-f003:**
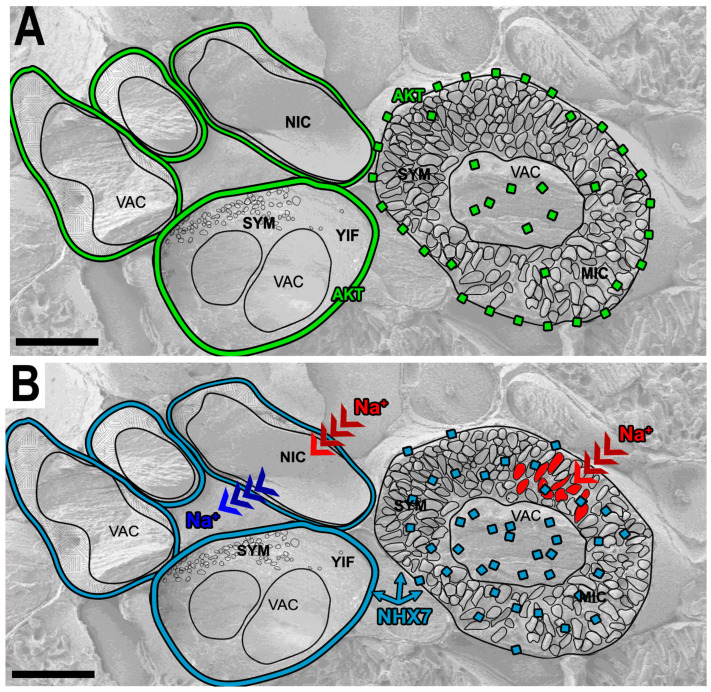
(**A**,**B**). Scheme of the distribution of the potassium channel AKT1 protein (**A**) and ion Na^+^/K^+^ exchanger NHX7 protein (**B**) in infected and non-infected cells of a root nodule. (**A**,**B**) Both transporters in young infected(YIC) and non-infected cells (NIC) are located on the plasma membrane; membranes are labeled in green for AKT1 and in blue for NHX7. In mature infected cells (MIC), AKT1 and NHX7 proteins are shed from the plasma membrane (green dotted signal and blue dotted signal, respectively) and transferred to the host cell vacuole (Vac). Due to the dislocation of the NHX7 Na^+^/K^+^ ion exchanger from the plasma membrane of the infected cell under the application of salt, Na^+^ ions (red signal, red arrows) are transferred and accumulated in the infected cell and in symbiosomes (marked in red). In non-infected cells, Na^+^ ions are transferred in and out of the plant cell (red and blue arrows) as the distribution of NHX7 on plasma membrane of non-infected cell is maintained correctly. Abbreviations: NIC—non-infected cell, YIC—young infected cell, MIC—mature infected cell, VAC—vacuole, T—tonoplast, PM—plasma membrane, IT—infection thread, SYM—symbiosomes. Bar: 12.5 μm.

## Data Availability

The literature data used in the Review are available as publications in relevant scientific journals.

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
