# Peer review of "Rapid Changes to Endomembrane System of Infected Root Nodule Cells to Adapt to Unusual Lifestyle"

_ijms, 2023, doi:10.3390/ijms24054647_

Round 1

Reviewer 1 Report

In the manuscript entitled ‘Endomembrane system of root nodule infected cell: successes and failures’ the author summarizes the cell biological changes that nodule cells undergo in order to accommodate rhizobia intracellularly. These include membrane synthesis and repurposing, cytoskeleton rearrangement, cell expansion, and vacuole defunctionalisation. Intracellular accommodation is a fascinating aspect of the root nodule symbiosis, and arguably, one of the most unique steps of this symbiosis. Nevertheless, molecular insights into this process are still lacking. Although the author does a good job at reviewing past and current literature on the topic and in shedding light into this underexplored yet key step of the root nodule symbiosis, the manuscript would benefit from clearer writing. Below you can find detailed comments. Examples listed below are intended as examples and are not complete list of all issues in the text. Please revise the whole article based on the comments.

Major comments:

1.       The term rhizobia should not be confused with bacteria from the genus Rhizobium. Rhizobia is a broader term, which comprises a paraphyletic group of gram-negative bacteria (mostly alphaproteobateria but also some betaproteobateria) that establish root nodule symbiosis with legume plants. Rhizobium is only one of many other rhizobia genera. Please correct in the text (e.g. lines 7, 20).

2.       Please clarify the following sentences:

a.       Line 53:’colonies enter the apoplast of nodule primordia’. Do you mean the intercellular space or the lumen of the IT?

b.       Line 56:’probably due to the developed cell wall and a turgor pressure are out of reach’.

c.       Lines 56-57: ‘The entrance of microsymbionts triggers the formation of specific asymmetric protrusions of plasma membrane’. What do you mean with entrance? Do you mean that the entry of the bacteria causes the formation of IT, like a physical force? And not perception of a bacteria signal?

d.       Lines 134-136: ‘The infected cells are growing concomitantly with increasing symbiosome population. As a result, the infected cell’s volume increases several times in comparison with non-infected ones’. In this sentence you imply that the cell volume increase is caused by the increase in symbiosome numbers. How do you establish causality? Where is the empirical evidence?

e.       Lines 238-240: The tonoplast aquaporin TIP1g is retargeted to the symbiosome membrane that gives to microsymbiont the advantage for water transport from the host cell to bacteroid. What is the evidence that water is transported in that direction?

f.        Line 262: ‘…infected cell death after 4-5 weeks of existence’. Is this true for all legumes?

g.       Line 283: why hypoxia would promote deterioration of ion exchange?

3.       Please cite original work instead of other reviews when possible. E.g.:

a.       Lines 57-60: references 17, 18, 19.

b.       Line 74: reference 28.

4.       Citations missing. E.g.:

a.       The entrance of microsymbionts triggers the formation of specific asymmetric protrusions of plasma membrane.

b.       The growth of symbiotic structures in infected cells is mainly isodiametric for unwalled droplets and symbiosomes, apart of a tip growth of infection threads.

c.       short lifespan of symbiotic cells.

d.       symbiosomes with their ”fake“ identity of “late endosomes/young vacuoles” due to the presence of endocytotic identical markers on the membrane, are able to survive and fix atmospheric nitrogen for several days, the time to fulfil the task for infected cell.

e.       genetically identical uninfected cell can live for years

5.       The review would profit of figures illustrating the main concepts. In the current version there is only one figure illustrating one aspect of the review, which is not even a central message.

Minor comments:

1.       I recommend to place the citations directly after the corresponding sentence and not at the end of the paragraph. It is easier for the reader.

2.       Lines 28, 53: a colony is defined as a group of bacteria derived from the same mother cell. Is it known if the bacteria infecting a plant cell all come from the same mother bacterial cell?

3.       Line 44: plural of locus is loci not locuses.

4.       Spelling or grammar mistakes. E.g.:

a.       Line 56: reduced instead of reduces.

b.       Line 198, 204. M. truncatula, A. thaliana. Space missing. Arabidopsis thaliana should be spelled fully when first mentioned.

c.       Lines 212-214: remove that.

5.       Please clarify when a finding is specific to a legume species (e.g. Medicago truncatula) or when it is applicable for all or a variety of legumes.

6.       References format.

Author Response

A: I am very thankful for Reviewer1 for the comments and suggestions. I have accepted all corrections that have been recommended by Reviewer1. The title of review has been changed according to Reviewers suggestions. Two additional schemes are added (Fig.1,2) that are illustrating the change in symbiosome membrane identity and pH of infected cells vacuoles.

Rev 1

Major comments:

  1. The term rhizobia should not be confused with bacteria from the genus Rhizobium. Rhizobia is a broader term, which comprises a paraphyletic group of gram-negative bacteria (mostly alphaproteobateria but also some betaproteobateria) that establish root nodule symbiosis with legume plants. Rhizobium is only one of many other rhizobia genera. Please correct in the text (e.g. lines 7, 20).

A: term is corrected

  1. Please clarify the following sentences:
  2. Line 53:’colonies enter the apoplast of nodule primordia’. Do you mean the intercellular space or the lumen of the IT?

A: I have changed “apoplast” to “intercellular space”

  1. Line 56:’probably due to the developed cell wall and a turgor pressure are out of reach’.

I've adjusted the phrase, line 54-57. The meaning was that mature cells with developed vacuole are out of reach for rhizobia.

  1. Lines 56-57: ‘The entrance of microsymbionts triggers the formation of specific asymmetric protrusions of plasma membrane’. What do you mean with entrance? Do you mean that the entry of the bacteria causes the formation of IT, like a physical force? And not perception of a bacteria signal?

A:The paragraph 54-57 is corrected.

      The perception of bacteria signal is, for sure, is quite important. It is well established that it  is causing the changes in hormonal balance that leads to the induction of a new meristem, it helps with the entrance of rhizobia to the intracellular space, and it induces the expression of genes, necessary for symbiosis development as well as new membrane nanodomains, remorin, flotilins ets. And yes, proliferating bacteria exert also a physical pressure, host cell is perceived it as a touch and the genes involved in touch response like synaptotagmins are expressed in young infected host cell (Gavrin et al., 2017, Frontiers PS).

  1. Lines 134-136: ‘The infected cells are growing concomitantly with increasing symbiosome population. As a result, the infected cell’s volume increases several times in comparison with non-infected ones’. In this sentence you imply that the cell volume increase is caused by the increase in symbiosome numbers. How do you establish causality? Where is the empirical evidence?

A: We quantitively estimated volumes of the cells, vacuoles and bacteria using GFP-tagged proteins and 3D reconstruction of Z-stacks obtained by confocal microscopy (Gavrin et al., 2014, The Plant Cell). Yes, the volume of infected cell linearly increases following the increase in symbiosome numbers/ volume. The volume of host cell vacuole was reduced 4-fold in cell layer next to interzone, but the volume of host cell still increase because of the mechanical pressure of proliferating and growing symbiosomes.  

  1. Lines 238-240: The tonoplast aquaporin TIP1g is retargeted to the symbiosome membrane that gives to microsymbiont the advantage for water transport from the host cell to bacteroid. What is the evidence that water is transported in that direction?

A: The functions of TIP1 as an aquaporin has been tested in Xenopus laevis oocytes. The functional of TIP1g aquaporin in nodule development has been tested with RNAi experiments. The nodules with high levels of silencing of TIP1 (95%) are Fix-. In these nodules the symbiosomes are small, they do not grow and do not mature. (Gavrin et al., 2014, The Plant Cell).

        Line 262: ‘…infected cell death after 4-5 weeks of existence’. Is this true for all legumes?

A: I cannot tell for all legumes. But in last 35 years I have been working with the nodules of pea, beans, soybeans, alfalfa, lupins, and nodules of non-legume tree Parasponia induced by rhizobia, as well as with different mutants of hosts and bacteria. It is true for infected cells of all these species, the starts of senescence can be detected on 35-40 days. The nodule can be attached to the root, but the functional living zone contains up to 32-36 cell layers, and the basal part contains dead or dying cells that are repopulated by saprophytic rhizobia or microbiota of other species plus vascular bundles.  A classical work of C Timmers , E Soupène, M C Auriac, F de Billy, J Vasse, P Boistard, G Truchet. Saprophytic intracellular rhizobia in alfalfa nodules. Mol Plant Microbe Interact.  2000 Nov;13(11):1204-13. doi: 10.1094/MPMI.2000.13.11.1204 contains the typical images.

.       Line 283: why hypoxia would promote deterioration of ion exchange?

Ion pumping via ion channels is energetically expensive process, and energy deficiency is leads to generalized «channel arrest» because hypoxia affects ATP supply/demands. Since the discovery of oxygen-dependent potassium channels in 1988, thousands of papers have been devoted to this topic, mostly they have been studying animal models. It is understandable, hypoxia is interlinked with clinical conditions of ischemia. Usually, the most affected was an inward transport of potassium, channel-dependent. In animal cells it causes the deterioration of Na+\K+ level, and affects also other intermembrane transport. In plants, the studies in the field of oxygen deficiency due to waterlogging have shown the defects in ion transport. The research concerning the study of hypoxia, for example, in the group of Shabala, that have studied the effects of hypoxia and salinity, have shown that the most negatively affected was a potassium transport that caused the drop in K/Na level. The putative negative effect of hypoxia for root nodule infected cells was not under consideration. The common believe is that hypoxic conditions are supporting the activity of nitrogenase hence it is beneficial for symbiosis. The expression of some ion channels in root nodule infected zone is downregulated (Trifonova et al., 2022).

  1. Please cite original work instead of other reviews when possible.

A:I agree with Reviewer opinion concerning the citation. But in the Review article there is a word limit. The review as it was submitted was only 9 words left to be out of word limit, and I am afraid that now the limit already is exceeded after the correction made to the text according to the advices of the reviewers. Even the brief description of Nod-factor pathway with links to the work of colleagues who discovered each gene, will takes 150-200 words. Due to this I ask the Reviewer to allow me to leave the references to excellent reviews of my colleagues.

  1. Lines 57-60: references 17, 18, 19.
  2. Line 74: reference 28.
  3. Citations missing. E.g.:

A The reference 4 is cited (line 26)

  1. The entrance of microsymbionts triggers the formation of specific asymmetric protrusions of plasma membrane.
  2. The growth of symbiotic structures in infected cells is mainly isodiametric for unwalled droplets and symbiosomes, apart of a tip growth of infection threads.
  3. short lifespan of symbiotic cells.
  4. symbiosomes with their ”fake“ identity of “late endosomes/young vacuoles” due to the presence of endocytotic identical markers on the membrane, are able to survive and fix atmospheric nitrogen for several days, the time to fulfil the task for infected cell.
  5. genetically identical uninfected cell can live for years

A:These statements, suggested by Reviewer1, have been included to the Conclusion lines 337-344

  1. The review would profit of figures illustrating the main concepts. In the current version there is only one figure illustrating one aspect of the review, which is not even a central message.

A: additional two schemes are added (Fig.1,2) that are illustrating the change in symbiosome membrane identity and pH of infected cells vacuoles. Figure legend is added to the text.

Minor comments:

  1. I recommend to place the citations directly after the corresponding sentence and not at the end of the paragraph. It is easier for the reader.

A: I have made it according to the model given by the journal

  1. Lines 28, 53: a colony is defined as a group of bacteria derived from the same mother cell. Is it known if the bacteria infecting a plant cell all come from the same mother bacterial cell?

A  Common believe is that the bacteria, entrapped in root hair hook, is a single progenitor for all bacteria in the nodule. However exists the data showing the infection threads carrying mixed bacteria, like GFP and RFP-tagged (D.J. Gage , Infection and Invasion of Roots by Symbiotic, Nitrogen-Fixing Rhizobia during Nodulation of Temperate Legumesdoi: 10.1128/MMBR.68.2.280-300.2004, Microbiol Mol Biol Rev. 2004 68(2): 280–300.).

 Line 44: plural of locus is loci not locuses.

A: corrected

  1. Spelling or grammar mistakes. E.g.:
  2. Line 56: reduced instead of reduces.

A:Corrected

  1. Line 198, 204. M. truncatulaA. thaliana. Space missing. Arabidopsis thalianashould be spelled fully when first mentioned.

A:Corrected

  1. Lines 212-214: remove that.

A: the phrase is corrected

  1. Please clarify when a finding is specific to a legume species (e.g. Medicago truncatula) or when it is applicable for all or a variety of legumes.

Corrected,lines 148, 196,210,250

  1. References format.

Reviewer 2 Report

Legume-rhizobia symbiosis has been studying for many years and the initials steps of the root nodule development are becoming to be clearer. But the processes that take place in a mature nodule are still hidden. This minireview summarizes the information about the endomembrane system of root nodule infected cell, discuses the cost of this symbiosis and asks a question about the reason of termination of root nodule lifespan in a certain period of time.

Reading this review, it is difficult to understand is this information appropriate for all types of nodules or for a certain type? Maybe it is better somewhere to note that Medicago truncatula nodules are better studied and this information is highly likely correct for indertminate nodules. Or only for annual nodules. Because perennial nodules also exist and they do not terminate after 6-7 weeks of lifespan.

Also speaking about the description of indeterminate nodules it is better to give some examples of species (line 41). It  is good to do the same for determinate and lupinoid types of nodules. Moreover part 3 deals with M. truncatula. And it is not clear what the type of nodules is this and why only M. truncatula is mentioned? Lotus japonicus is well studied too…

Line 42-43 – ‘cell layers are develop between’ the gramma should be checked.

Lines 53-54 – this part is not clear…what do you mean  ‘initial stages’? the stage after bacteria entered root cortex?

Line 54 - it should be ‘enter the host cell’

Line 68 - 69 - according to this information readers might suppose that NCR peptides were found only in M. truncatula. But NCR peptides were found in different species of IRLC clade and M. truncatula belongs to this clade.

Line 154 – it should be ‘at the same time’

Line 172 – check the grammar. It should be ‘ER is unexplored yet’

Line 204 – this is the first time of A. thaliana, so it should be full genus name.

Lines 212-215- either something is missed or ‘and’ has to be deleted.

Line 276 – check spaces between words.

Line 285 – there should be either may or is

Line 298 – check the grammar

Figure 1. definitions should be given for NIC, VAC, SYM, YIF, MIC. At B at right cell VAC should be replaced a little.

Line 321 – check spaces.

Line 346 – it should be ‘the works aimed to improve certain aspects’

Author Response

A: I am very thankful for Reviewer2 for the comments and suggestions. I have accepted all corrections that have been recommended by Reviewer.  Two additional schemes are added (Fig.1,2) that are illustrating the change in symbiosome membrane identity and pH of infected cells vacuoles.

Legume-rhizobia symbiosis has been studying for many years and the initials steps of the root nodule development are becoming to be clearer. But the processes that take place in a mature nodule are still hidden. This minireview summarizes the information about the endomembrane system of root nodule infected cell, discuses the cost of this symbiosis and asks a question about the reason of termination of root nodule lifespan in a certain period of time.

Reading this review, it is difficult to understand is this information appropriate for all types of nodules or for a certain type? Maybe it is better somewhere to note that Medicago truncatula nodules are better studied and this information is highly likely correct for indertminate nodules. Or only for annual nodules. Because perennial nodules also exist and they do not terminate after 6-7 weeks of lifespan.

Also speaking about the description of indeterminate nodules it is better to give some examples of species (line 41). It  is good to do the same for determinate and lupinoid types of nodules.

A: corrected, examples of species are added, lines 46-50

 Moreover part 3 deals with M. truncatula. And it is not clear what the type of nodules is this and why only M. truncatula is mentioned? Lotus japonicus is well studied too…

A: The references to works specified works used M.truncatula have been added (lines 87,109, 148, 196,210,250).

Line 42-43 – ‘cell layers are develop between’ the gramma should be checked.

A: the phrase is corrected

Lines 53-54 – this part is not clear…what do you mean  ‘initial stages’? the stage after bacteria entered root cortex?

Line 54 - it should be ‘enter the host cell’

A:The paragraph is corrected, lines 55-60.

Line 68 - 69 - according to this information readers might suppose that NCR peptides were found only in M. truncatula. But NCR peptides were found in different species of IRLC clade and M. truncatula belongs to this clade.

A:Corrected, lines 72-73

Line 154 – it should be ‘at the same time’

A: Corrected, line 170

Line 172 – check the grammar. It should be ‘ER is unexplored yet’

A:corrected line 189

Line 204 – this is the first time of A. thaliana, so it should be full genus name.

A:Corrected, line 221

Lines 212-215- either something is missed or ‘and’ has to be deleted.

A:corrected,lines 229-230

Line 276 – check spaces between words.

A:Corrected line 294

Line 285 – there should be either may or is

Corrected line 310

Line 298 – check the grammar

Corrected, line 303

Figure 1. definitions should be given for NIC, VAC, SYM, YIF, MIC. At B at right cell VAC should be replaced a little.

A:Image is corrected. The Figure legend in corrected and include now 3 figures :Fig.1A, Fig.2 A,B. Definitions for abbreviation NIC, VAC, SYM, YIF, MIC. are included to the Figure legend

Line 321 – check spaces.

A:Corrected, lines 328-329

Line 346 – it should be ‘the works aimed to improve certain aspects’

A:Corrected lines 360-361

Reviewer 3 Report

The review article entitled "Endomembrane system of root nodule infected cell: success and failures" is comprehensive and well organized, so it can be accepted after minor revision.

Title: I recommend deleting the last part of the title "success and failures" because it is not clear who's success and failures, a plant, rhizobia, or scientists.

I also recommend adding some model figures of symbiotic processes and the characteristics of symbiosome in addition to Figure 1. Figure 1 shows only K+ transport.

The English editing is necessary, for example at line 22 synthetize is synthesize,

Author Response

A: I am very thankful for Reviewer3 for the comments and suggestions. I have accepted all corrections that have been recommended by Reviewer. The title of review has been changed according to Reviewer suggestions. Two additional schemes are added (Fig.1,2) that are illustrating the change in symbiosome membrane identity and pH of infected cells vacuoles.

The review article entitled "Endomembrane system of root nodule infected cell: success and failures" is comprehensive and well organized, so it can be accepted after minor revision.

Title: I recommend deleting the last part of the title "success and failures" because it is not clear who's success and failures, a plant, rhizobia, or scientists.

A:The title is corrected, "success and failures"  are deleted.

I also recommend adding some model figures of symbiotic processes and the characteristics of symbiosome in addition to Figure 1. Figure 1 shows only K+ transport.

A:Two more figures have been added that illustrate the change in symbiosome membrane identity and  modifications in pH of infected cells vacuoles. Figure legend is added to the text.

The English editing is necessary, for example at line 22 synthetize is synthesize,

A Corrected, line 22

The Ms has been red by native English speaking person.